# DSD: DENSE-SPARSE-DENSE TRAINING FOR DEEP NEURAL NETWORKS

**Song Han**[*]**, Huizi Mao, Enhao Gong, Shijian Tang, William J. Dally**[†]
Stanford University
`{songhan,huizi,enhaog,sjtang,dally}@stanford.edu`

**Jeff Pool**[*]**, John Tran, Bryan Catanzaro**
NVIDIA
`{jpool,johntran,bcatanzaro}@nvidia.com`

**Sharan Narang**[*]**, Erich Elsen**[‡]
Baidu Research
`sharan@baidu.com`

**Peter Vajda, Manohar Paluri**
Facebook
`{vajdap,mano}@fb.com`

## ABSTRACT

Modern deep neural networks have a large number of parameters, making them very hard to train. We propose DSD, a dense-sparse-dense training flow, for regularizing deep neural networks and achieving better optimization performance. In the first D (Dense) step, we train a dense network to learn connection weights and importance. In the S (Sparse) step, we regularize the network by pruning the unimportant connections with small weights and retraining the network given the sparsity constraint. In the final D (re-Dense) step, we increase the model capacity by removing the sparsity constraint, re-initialize the pruned parameters from zero and retrain the whole dense network. Experiments show that DSD training can improve the performance for a wide range of CNNs, RNNs and LSTMs on the tasks of image classification, caption generation and speech recognition. On ImageNet, DSD improved the Top1 accuracy of GoogLeNet by 1.1%, VGG-16 by 4.3%, ResNet-18 by 1.2% and ResNet-50 by 1.1%, respectively. On the WSJ'93 dataset, DSD improved DeepSpeech and DeepSpeech2 WER by 2.0% and 1.1%. On the Flickr-8K dataset, DSD improved the NeuralTalk BLEU score by over 1.7. DSD is easy to use in practice: at training time, DSD incurs only one extra hyper-parameter: the sparsity ratio in the S step. At testing time, DSD doesn't change the network architecture or incur any inference overhead. The consistent and significant performance gain of DSD experiments shows the inadequacy of the current training methods for finding the best local optimum, while DSD effectively achieves superior optimization performance for finding a better solution. DSD models are available to download at https://songhan.github.io/DSD.

## 1 INTRODUCTION

Deep neural networks (DNNs) have shown significant improvements in many application domains, ranging from computer vision (He et al. (2015)) to natural language processing (Luong et al. (2015)) and speech recognition (Amodei et al. (2015)). The abundance of powerful hardware makes it easier to train complicated DNN models with large capacities. The upside of complicated models is that they are very expressive and can capture the highly non-linear relationship between features and output. The downside of such large models is that they are prone to capturing the noise, rather than the intended pattern, in the training dataset. This noise does not generalize to new datasets, leading to over-fitting and a high variance.

---

[*]Indicates equal contribution
[†]Also at NVIDIA
[‡]Now at Google Brain. eriche@google.com

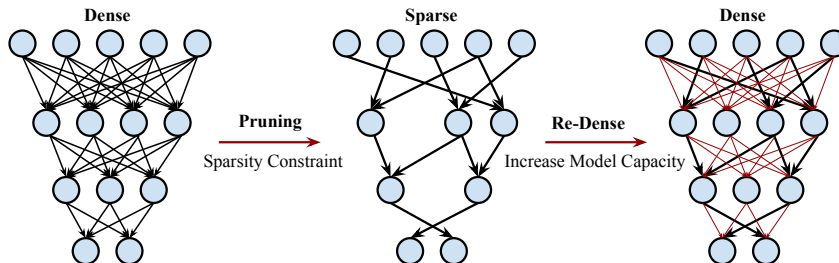

Figure 1: Dense-Sparse-Dense Training Flow. The sparse training regularizes the model, and the final dense training restores the pruned weights (red), increasing the model capacity without overfitting.

---

**Algorithm 1:** Workflow of DSD training

---

**Initialization:** $W^{(0)}$ $with$ $W^{(0)} \sim N(0, \Sigma)$
**Output :** $W^{(t)}$.

―――――――――――――――――――――― *Initial Dense Phase* ――――――――――――――――――――――

**while** *not converged* **do**
 $W^{(t)} = W^{(t-1)} - \eta^{(t)} \nabla f(W^{(t-1)}; x^{(t-1)})$;
 $t = t + 1$;
**end**

―――――――――――――――――――――― *Sparse Phase* ――――――――――――――――――――――-

// *initialize the mask by sorting and keeping the Top-k weights.*
$S = sort(|W^{(t-1)}|)$; $\lambda = S_{k_i}$; $Mask = \mathbb{1}(|W^{(t-1)}| > \lambda)$;
**while** *not converged* **do**
 $W^{(t)} = W^{(t-1)} - \eta^{(t)} \nabla f(W^{(t-1)}; x^{(t-1)})$;
 $W^{(t)} = W^{(t)} \cdot Mask$;
 $t = t + 1$;
**end**

―――――――――――――――――――――― *Final Dense Phase* ――――――――――――――――――――――

**while** *not converged* **do**
 $W^{(t)} = W^{(t-1)} - \eta^{(t)} \nabla f(W^{(t-1)}; x^{(t-1)})$;
 $t = t + 1$;
**end**
**goto** *Sparse Phase* for iterative DSD;

---

In contrast, simply reducing the model capacity would lead to the other extreme, causing a machine learning system to miss the relevant relationships between features and target outputs, leading to under-fitting and a high bias. Bias and variance are hard to optimize at the same time.

To solve this problem, we propose a dense-sparse-dense training flow (DSD), a novel training strategy that starts from a dense model from conventional training, then regularizes the model with sparsity-constrained optimization, and finally increases the model capacity by restoring and retraining the pruned weights. At testing time, the final model produced by DSD still has the same architecture and dimension as the original dense model, and DSD training doesn't incur any inference overhead. We experimented DSD training on 7 mainstream CNN / RNN / LSTMs and found consistent performance gains over its comparable counterpart for image classification, image captioning and speech recognition.

## 2  DSD TRAINING FLOW

Our DSD training employs a three-step process: dense, sparse, re-dense. Each step is illustrated in Figure 1 and Algorithm 1. The progression of weight distribution is plotted in Figure 2.

**Initial Dense Training:** The first D step learns the connection weights and importance via normal network training on the dense network. Unlike conventional training, however, the goal of this D step is not only to learn the values of the weights; we are also learning which connections are important. We use a simple heuristic to quantify the importance of the weights using their absolute value.

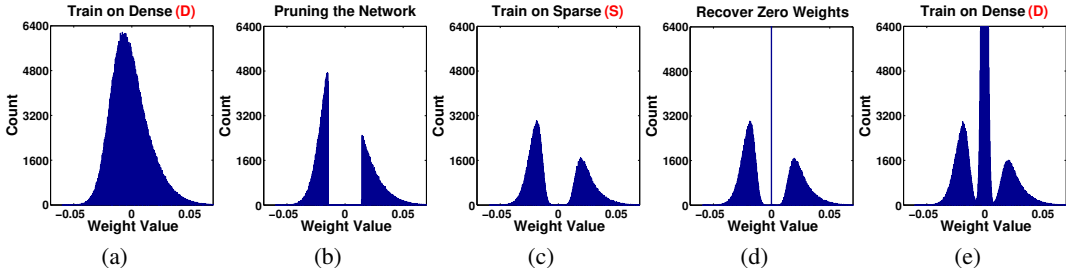

Figure 2: Weight distribution of a layer of GoogLeNet at different points in DSD training: the original GoogLeNet (a), pruned (b), after retraining with the sparsity constraint (c), ignoring the sparisty constraint and recovering the zero weights (d), and after retraining the dense network (e).

**Sparse Training:** The S step prunes the low-weight connections and trains a sparse network. We applied the *same* sparsity to all the layers, thus there's a single hyper parameter: the *sparsity*, the percentage of weights that are pruned to 0. For each layer $W$ with $N$ parameters, we sorted the parameters, picked the k-th largest one $\lambda = S_k$ as the threshold where $k = N * (1 - sparsity)$, and generated a binary mask to remove all the weights smaller than $\lambda$. Details are shown in Algorithm 1 .

We remove small weights because of the Taylor expansion. The loss function and its Taylor expansion are shown in Equation (1)(2). We want to minimize the increase in $Loss$ when conducting a hard thresholding on the weights, so we need to minimize the first and second terms in Equation 2. Since we are zeroing out parameters, $\Delta W_i$ is actually $W_i - 0 = W_i$. At the local minimum where $\partial Loss / \partial W_i \approx 0$ and $\frac{\partial^2 Loss}{\partial W_i^2} > 0$, only the second order term matters. Since second order gradient $\partial^2 Loss / \partial W_i^2$ is expensive to calculate and $W_i$ has a power of 2, we use $|W_i|$ as the metric of pruning. Smaller $|W_i|$ means a smaller increase to the loss function.

$$Loss = f(x, W_1, W_2, W_3...) \tag{1}$$

$$\Delta Loss = \frac{\partial Loss}{\partial W_i} \Delta W_i + \frac{1}{2} \frac{\partial^2 Loss}{\partial W_i^2} \Delta W_i{}^2 + ... \tag{2}$$

Retraining while enforcing the binary mask in each iteration, we converted a dense network into a sparse network that has a known sparsity support and can fully recover or even increase the original accuracy of initial dense model under the sparsity constraint. The *sparsity* is the same for all the layers and can be tuned using validation. We find a sparsity value between 25% and 50% generally works well in our experiments.

**Final Dense Training:** The final D step recovers the pruned connections, making the network dense again. These previously-pruned connections are initialized to zero and the entire network is retrained with 1/10 the original learning rate (since the sparse network is already at a good local minima). Hyper parameters like dropout ratios and weight decay remained unchanged. By restoring the pruned connections, the final D step increases the model capacity of the network and makes it possible to arrive at a better local minima compared with the sparse model from the S step.

To visualize the DSD training flow, we plotted the progression of the weight distribution in Figure 2. The figure is plotted using GoogLeNet's inception_5b3x3 layer, and we found this progression of weight distribution very representative for VGGNet and ResNet as well. The original distribution of weight is centered on zero with tails dropping off quickly. Pruning is based on absolute value so after pruning the large center region is truncated away. The un-pruned network parameters adjust themselves during the retraining phase, so in (c), the boundary becomes soft and forms a bimodal distribution. In (d), at the beginning of the re-dense training step, all the pruned weights come back again and are reinitialized to zero. Finally, in (e), the pruned weights are retrained together with the un-pruned weights. In this step, we kept the same learning hyper-parameters (weight decay, learning rate, etc.) for pruned weights and un-pruned weights. Comparing Figure (d) and (e), the un-pruned weights' distribution almost remained the same, while the pruned weights became distributed further around zero. The overall mean absolute value of the weight distribution is much smaller. This is a good phenomenon: choosing the *smallest* vector that solves the learning problem suppresses irrelevant components of the weight vector ( Moody et al. (1995)).

Table 1: Overview of the neural networks, data sets and performance improvements from DSD.

| Neural Network | Domain | Dataset | Type | Baseline | DSD | Abs. Imp. | Rel. Imp. |
|---|---|---|---|---|---|---|---|
| GoogLeNet | Vision | ImageNet | CNN | 31.1%[1] | **30.0%** | 1.1% | 3.6% |
| VGG-16 | Vision | ImageNet | CNN | 31.5%[1] | **27.2%** | 4.3% | 13.7% |
| ResNet-18 | Vision | ImageNet | CNN | 30.4%[1] | **29.2%** | 1.2% | 4.1% |
| ResNet-50 | Vision | ImageNet | CNN | 24.0%[1] | **22.9%** | 1.1% | 4.6% |
| NeuralTalk | Caption | Flickr-8K | LSTM | 16.8[2] | **18.5** | 1.7 | 10.1% |
| DeepSpeech | Speech | WSJ'93 | RNN | 33.6%[3] | **31.6%** | 2.0% | 5.8% |
| DeepSpeech-2 | Speech | WSJ'93 | RNN | 14.5%[3] | **13.4%** | 1.1% | 7.4% |

[1] Top-1 error. VGG/GoogLeNet baselines from the Caffe Model Zoo, ResNet from Facebook.
[2] BLEU score baseline from Neural Talk model zoo, the higher the better.
[3] Word error rate: DeepSpeech2 is trained with a portion of Baidu internal dataset with only max decoding to show the effect of DNN improvement.

## 3 RELATED WORK

**Dropout and DropConnect:** DSD, Dropout (Srivastava et al. (2014)) and DropConnnect (Wan et al. (2013)) can all regularize neural networks and prevent over-fitting. The difference is that Dropout and DropConnect use a *random* sparsity pattern at each SGD iteration, while DSD training learns with a *deterministic* data driven sparsity pattern throughout sparse training. Our experiments on VGG16, GoogLeNet and NeuralTalk show that DSD training can work together with Dropout.

**Distillation:** Model distillation (Hinton et al. (2015)) is a method that can transfer the learned knowledge from a large model to a small model, which is more efficient for deployment. This is another method that allows for performance improvements in neural networks without architectural changes.

**Model Compression:** Both model compression (Han et al. (2016; 2015)) and DSD training use network pruning (LeCun et al. (1990); Hassibi et al. (1993)). The difference is that the focus of DSD training goes beyond maintaining the accuracy. DSD is able to further improve the accuracy by considerable margins. Another difference is that DSD training doesn't require aggressive pruning. A modestly pruned network (50%-60% sparse) can work well. However, model compression requires aggressively pruning the network to achieve high compression rates.

**Sparsity Regularization and Hard Thresholding:** the truncation-based sparse network has been theoretically analyzed for learning a broad range of statistical models in high dimensions (Langford et al. (2009); Yuan & Zhang (2013); Wang et al. (2014)). A similar training strategy with iterative hard thresholding and connection restoration is proposed by Jin et al. (2016) during the same time period as, but independently from, DSD. Sparsity regularized optimization is heavily applied in Compressed Sensing (Candes & Romberg (2007)) to find optimal solutions to the inverse problems in highly under-determined systems based on the sparsity assumption.

## 4 EXPERIMENTS

We applied DSD training to different kinds of neural networks in different domains. We found that DSD training improved the accuracy for *all* these networks compared to the baseline networks that were not trained with DSD. The neural networks are chosen from CNN, RNN and LSTMs; the datasets covered image classification, speech recognition, and caption generation. For networks trained for ImageNet, we focus on GoogLeNet, VGG and ResNet, which are widely used in research and production. An overview of the networks, dataset and accuracy results are shown in Table 1. For the convolutional networks, we do not prune the first layer during the sparse phase, since it has only 3 channels and is very sensitive to pruning. The sparsity is the *same* for all the other layers, including convolutional and fully-connected layers. We do not change any other training hyper-parameters, and the initial learning rate at each stage is decayed the same as conventional training. The epochs are decided by when the loss converges. When the loss no longer decreases, we stop the training.

### 4.1 GOOGLENET

We experimented with the BVLC GoogLeNet (Szegedy et al. (2015)) model obtained from the Caffe Model Zoo (Jia (2013)). It has 13 million parameters and 57 convolutional layers. We pruned each layer (except the first) to 30% sparsity. Retraining the sparse network gave some improvement in accuracy due to regularization, as shown in Table 2. After the final dense training step, GoogLeNet's error rates were reduced by 1.12% (Top-1) and 0.62% (Top-5) over the baseline.

We compared DSD v.s. conventional training for the *same number of epochs* by dropping the learning rate upon "convergence" and continuing to learn. The result is shown as LLR (lower the learning rate). The training epochs for LLR is equal to that of Sparse+re-Dense as a fair comparison. LLR can not achieve the same accuracy as DSD.

Table 2: DSD results on GoogLeNet

| GoogLeNet | Top-1 Err | Top-5 Err | Sparsity | Epochs | LR |
|---|---|---|---|---|---|
| Baseline | 31.14% | 10.96% | 0% | 250 | 1e-2 |
| Sparse | 30.58% | 10.58% | 30% | 11 | 1e-3 |
| DSD | **30.02%** | **10.34%** | 0% | 22 | 1e-4 |
| LLR | 30.20% | 10.41% | 0% | 33 | 1e-5 |
| Improve (abs) | 1.12% | 0.62% | - | - | - |
| Improve (rel) | **3.6%** | **5.7%** | - | - | - |

### 4.2 VGGNET

We explored DSD training on VGG-16 (Simonyan & Zisserman (2014)), which is widely used in detection, segmentation and transfer learning. The baseline model is obtained from the Caffe Model Zoo (Jia (2013)). Similar to GoogLeNet, each layer is pruned to 30% sparsity. DSD training greatly reduced the error by 4.31% (Top-1) and 2.65% (Top-5), detailed in Table 3. DSD also wins over the LLR result by a large margin.

Table 3: DSD results on VGG-16

| VGG-16 | Top-1 Err | Top-5 Err | Sparsity | Epochs | LR |
|---|---|---|---|---|---|
| Baseline | 31.50% | 11.32% | 0% | 74 | 1e-2 |
| Sparse | 28.19% | 9.23% | 30% | 1.25 | 1e-4 |
| DSD | **27.19%** | **8.67%** | 0% | 18 | 1e-5 |
| LLR | 29.33% | 10.00% | 0% | 20 | 1e-7 |
| Improve (abs) | 4.31% | 2.65% | - | - | - |
| Improve (rel) | **13.7%** | **23.4%** | - | - | - |

### 4.3 RESNET

Deep Residual Networks (ResNets, He et al. (2015)) were the top performer in the 2015 ImageNet challenge. The baseline ResNet-18 and ResNet-50 models are provided by Facebook (2016). We prune to 30% sparsity uniformly, and a single DSD pass for these networks reduced top-1 error by 1.26% (ResNet-18) and 1.12% (ResNet-50), shown in Table 4. A second DSD iteration can further improve the accuracy. As a fair comparison, we continue train the original model by lowering the learning rate by another decade, but can't reach the same accuracy as DSD, as shown in the LLR row.

Table 4: DSD results on ResNet-18 and ResNet-50

| | ResNet-18 | | ResNet-50 | | | | |
|---|---|---|---|---|---|---|---|
| | Top-1 Err | Top-5 Err | Top-1 Err | Top-5 Err | Sparsity | Epochs | LR |
| Baseline | 30.43% | 10.76% | 24.01% | 7.02% | 0% | 90 | 1e-1 |
| Sparse | 30.15% | 10.56% | 23.55% | 6.88% | 30% | 45 | 1e-2 |
| DSD | **29.17%** | **10.13%** | **22.89%** | **6.47%** | 0% | 45 | 1e-3 |
| LLR | 30.04% | 10.49% | 23.58% | 6.84% | 0% | 90 | 1e-5 |
| Improve (abs) | 1.26% | 0.63% | 1.12% | 0.55% | - | - | - |
| Improve (rel) | **4.14%** | **5.86%** | **4.66%** | **7.83%** | - | - | - |

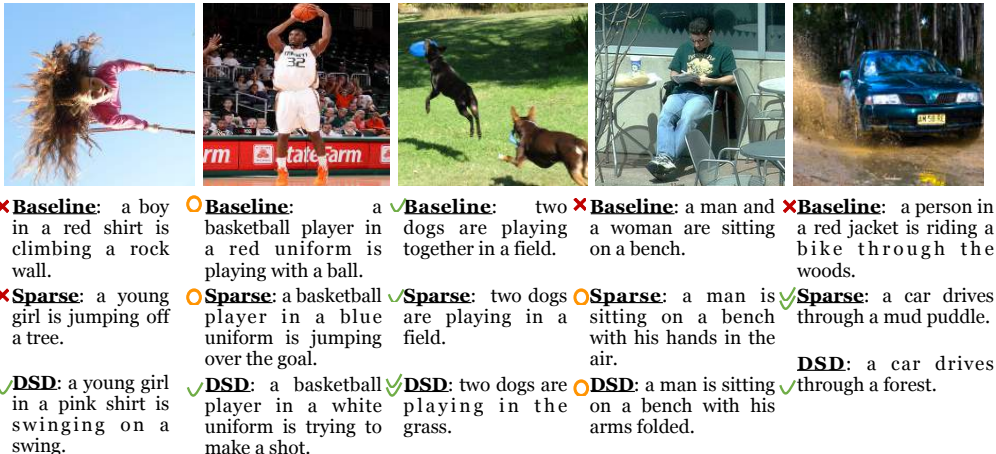

✗**Baseline**: a boy in a red shirt is climbing a rock wall.

○**Baseline**: a basketball player in a red uniform is playing with a ball.

✓**Baseline**: two dogs are playing together in a field.

✗**Baseline**: a man and a woman are sitting on a bench.

✗**Baseline**: a person in a red jacket is riding a bike through the woods.

✗**Sparse**: a young girl is jumping off a tree.

○**Sparse**: a basketball player in a blue uniform is jumping over the goal.

✓**Sparse**: two dogs are playing in a field.

○**Sparse**: a man is sitting on a bench with his hands in the air.

✓**Sparse**: a car drives through a mud puddle.

✓**DSD**: a young girl in a pink shirt is swinging on a swing.

✓**DSD**: a basketball player in a white uniform is trying to make a shot.

✓✓**DSD**: two dogs are playing in the grass.

○**DSD**: a man is sitting on a bench with his arms folded.

**DSD**: a car drives ✓through a forest.

Figure 3: Visualization of DSD training improving the performance of image captioning.

Table 5: DSD results on NeuralTalk

| NeuralTalk | BLEU-1 | BLEU-2 | BLEU-3 | BLEU-4 | Sparsity | Epochs | LR |
|---|---|---|---|---|---|---|---|
| Baseline | 57.2 | 38.6 | 25.4 | 16.8 | 0 | 19 | 1e-2 |
| Sparse | 58.4 | 39.7 | 26.3 | 17.5 | 80% | 10 | 1e-3 |
| DSD | **59.2** | **40.7** | **27.4** | **18.5** | 0 | 6 | 1e-4 |
| Improve(abs) | 2.0 | 2.1 | 2.0 | 1.7 | - | - | - |
| Improve(rel) | **3.5%** | **5.4%** | **7.9%** | **10.1%** | - | - | - |

## 4.4 NEURALTALK

We evaluated DSD training on RNN and LSTM beyond CNN. We applied DSD to NeuralTalk (Karpathy & Fei-Fei (2015)), an LSTM for generating image descriptions. It uses a CNN as an image feature extractor and an LSTM to generate captions. To verify DSD training on LSTMs, we fixed the CNN weights and only train the LSTM weights. The baseline NeuralTalk model we used is the flickr8k_cnn_lstm_v1.p downloaded from NeuralTalk Model Zoo.

In the pruning step, we pruned all layers except $W_s$, the word embedding lookup table, to 80% sparse. We used a higher sparsity than CNN's experiments based on the validation set of flickr8k. We retrained the remaining sparse network using the same weight decay and batch size as the original paper. The learning rate is tuned based on the validation set, shown in Table 5. Retraining the sparse network improved the BLUE score by [1.2, 1.1, 0.9, 0.7]. After getting rid of the sparsity constraint and retraining the dense network, the final results of DSD further improved the BLEU score by [2.0, 2.1, 2.0, 1.7] over baseline.

The BLEU score is not the sole criteria measuring auto-caption system. We visualized the captions generated by DSD training in Figure 3. In the first image, the baseline model mistakes the girl with a boy and the girl's hair with a rock wall; the sparse model can tell that it's a girl; and the DSD model can further identify the swing. In the the second image, DSD training can more accurately tell the player is in a white uniform and trying to make a shot, rather than the baseline just saying he's in a red uniform and playing with a ball. The performance of DSD training generalizes beyond these examples; more image caption results generated by DSD training are provided in the Appendix.

## 4.5 DEEPSPEECH

We explore DSD training on speech recognition tasks using both Deep Speech 1 (DS1) and Deep Speech 2 (DS2) networks (Hannun et al. (2014); Amodei et al. (2015)).

The DS1 model is a 5 layer network with 1 Bidirectional Recurrent layer, as described in Table 6. The training dataset used for this model is the Wall Street Journal (WSJ), which contains 81 hours of

Table 6: Deep Speech 1 Architecture

| Layer ID | 0 | 1 | 2 | 3 | 4 | 5 |
|---|---|---|---|---|---|---|
| Type | Conv | FC | FC | Bidirectional Recurrent | FC | CTCCost |
| #Params | 1814528 | 1049600 | 1049600 | 3146752 | 1049600 | 29725 |

Table 7: DSD results on Deep Speech 1: Word Error Rate (WER)

| DeepSpeech 1 | WSJ '92 | WSJ '93 | Sparsity | Epochs | LR |
|---|---|---|---|---|---|
| Dense Iter 0 | 29.82 | 34.57 | 0% | 50 | 8e-4 |
| Sparse Iter 1 | 27.90 | 32.99 | 50% | 50 | 5e-4 |
| Dense Iter 1 | 27.90 | 32.20 | 0% | 50 | 3e-4 |
| Sparse Iter 2 | 27.45 | 32.99 | 25% | 50 | 1e-4 |
| Dense Iter 2 | **27.45** | **31.59** | 0% | 50 | 3e-5 |
| Baseline | 28.03 | 33.55 | 0% | 150 | 8e-4 |
| Improve(abs) | 0.58 | 1.96 | - | - | - |
| Improve(rel) | **2.07%** | **5.84%** | - | - | - |

speech. The validation set consists of 1 hour of speech. The test sets are from WSJ'92 and WSJ'93 and contain 1 hour of speech combined. The Word Error Rate (WER) reported on the test sets for the baseline models is different from Amodei et al. (2015) due to two factors. First, in DeepSpeech2, the models were trained using much larger data sets containing approximately 12,000 hours of multi-speaker speech data. Secondly, WER was evaluated with beam search and a language model in DeepSpeech2; here the network output is obtained using only max decoding to show improvement in the neural network accuracy, and filtering out the other parts.

The first dense phase was trained for 50 epochs. In the sparse phase, weights are pruned in the Fully Connected layers and the Bidirectional Recurrent layer only (they are the majority of the weights). Each layer is pruned to achieve the same 50% sparsity and trained for 50 epochs. In the final dense phase, the pruned weights are initialized to zero and trained for another 50 epochs. For a fair comparison of baseline, we used Nesterov SGD to train, reduce the learning rate with each re-training, and keep all other hyper parameters unchanged. The learning rate is picked using our validation set.

We first wanted to compare the DSD results with a baseline model trained for the *same* number of epochs. The first 3 rows of Table 7 shows the WER when the DSD model is trained for 50+50+50=150 epochs, and the 6th line shows the baseline model trained by 150 epochs (the Same #Epochs as DSD). DSD training improves WER by 0.13 (WSJ '92) and 1.35 (WSJ '93) given the *same number of epochs* as the conventional training.

Given a second DSD iteration, accuracy can be further improved. In the second DSD iteration, each layer is pruned away 25% of the weights. Similar to the first iteration, the sparse model and subsequent dense model are further retrained for 50 epochs. The learning rate is scaled down for each re-training step. The results are shown in Table 7. Compared with the fully trained and converged baseline, the second DSD iteration improves WER by 0.58 (WSJ '92) and 1.96 (WSJ '93), a relative improvement of 2.07% (WSJ '92) and 5.84% (WSJ '93). So, we can do more DSD iterations (DSDSD) to further improve the performance. Adding more DSD iterations has a diminishing return.

## 4.6 DEEPSPEECH 2

To show how DSD works on deeper networks, we evaluated DSD on the Deep Speech 2 (DS2) network, described in Table 8. This network has 7 Bidirectional Recurrent layers with approximately 67 million parameters, around 8 times larger than the DS1 model. A subset of the internal English training set is used. The training set is comprised of 2,100 hours of speech. The validation set is comprised of 3.46 hours of speech. The test sets are from WSJ'92 and WSJ'93, which contain 1 hour of speech combined.

Table 9 shows the results of the two iterations of DSD training. For the first sparse re-training, similar to DS1, 50% of the parameters from the Bidirectional Recurrent Layers and Fully Connected

Table 8: Deep Speech 2 Architecture

| Layer ID | 0 | 1 | 2 | 3 - 8 | 9 | 10 |
|---|---|---|---|---|---|---|
| Type | 2DConv | 2DConv | BR | BR | FC | CTCCost |
| #Params | 19616 | 239168 | 8507840 | 9296320 | 3101120 | 95054 |

Table 9: DSD results on Deep Speech 2 (WER)

| DeepSpeech 2 | WSJ '92 | WSJ '93 | Sparsity | Epochs | LR |
|---|---|---|---|---|---|
| Dense Iter 0 | 11.83 | 17.42 | 0% | 20 | 3e-4 |
| Sparse Iter 1 | 10.65 | 14.84 | 50% | 20 | 3e-4 |
| Dense Iter 1 | 9.11 | 13.96 | 0% | 20 | 3e-5 |
| Sparse Iter 2 | 8.94 | 14.02 | 25% | 20 | 3e-5 |
| Dense Iter 2 | **9.02** | **13.44** | 0% | 20 | 6e-6 |
| Baseline | 9.55 | 14.52 | 0% | 60 | 3e-4 |
| Improve(abs) | 0.53 | 1.08 | - | - | - |
| Improve(rel) | **5.55%** | **7.44%** | - | - | - |

Layers are pruned. The Baseline model is trained for 60 epochs to provide a fair comparison with DSD training. The baseline model shows no improvement after 40 epochs. With one iteration of DSD training, WER improves by 0.44 (WSJ '92) and 0.56 (WSJ '93) compared to the fully trained baseline.

Here we show again that DSD can be applied multiple times or iteratively for further performance gain. A second iteration of DSD training achieves better accuracy as shown in Table 9. For the second sparse iteration, 25% of parameters in the Fully Connected layer and Bidirectional Recurrent layers are pruned. Overall DSD training achieves relative improvement of 5.55% (WSJ '92) and 7.44% (WSJ '93) on the DS2 architecture. These results are in line with DSD experiments on the smaller DS1 network. We can conclude that DSD re-training continues to show improvement in accuracy with larger layers and deeper networks.

## 5 DISCUSSION

Dense-Sparse-Dense training changes the optimization process and improves the optimization performance with significant margins by nudging the network with pruning and re-densifying. We conjecture that the following aspects contribute to the efficacy of DSD training.

**Escape Saddle Point:** Based on previous studies, one of the most profound difficulties of optimizing deep networks is the proliferation of saddle points (Dauphin et al. (2014)). Advanced optimization methods have been proposed to overcome saddle points. For a similar purpose but with a different approach, the proposed DSD method overcomes the saddle points by pruning and re-densifying framework. Pruning the converged model perturbs the learning dynamics and allows the network to jump away from saddle points, which gives the network a chance to converge at a better local or global minimum. This idea is also similar to Simulated Annealing ( Hwang (1988)). While Simulated Annealing randomly jumps with decreasing probability on the search graph, DSD deterministically deviates from the converged solution achieved in the first dense training phase by removing the small weights and enforcing a sparsity support. Similar to Simulated Annealing, which can escape sub-optimal solutions multiple times in the entire optimization process, DSD can also be applied iteratively to achieve further performance gains, as shown in the Deep Speech results.

**Significantly Better Minima:** After escaping saddle point, DSD achieved better minima. We measured both the training loss and validation loss, DSD training decreased the loss and error on both the training and the validation sets on ImageNet. We have also validated the significance of the improvements compared with conventional fine-tuning by t-test, shown in the appendix.

**Regularized and Sparse Training:** The sparsity regularization in the sparse training step moves the optimization to a lower-dimensional space where the loss surface is smoother and tend to be more robust to noise. More numerical experiments verified that both sparse training and the final DSD reduce the variance and lead to lower error (shown in the appendix).

**Robust re-initialization:** Weight initialization plays a big role in deep learning (Mishkin & Matas (2015)). Conventional training has only one chance of initialization. DSD gives the optimization a second (or more) chance during the training process to re-initialize from a more robust sparse training solution. We re-densify the network from the sparse solution which can be seen as a zero initialization for pruned weights. Other initialization methods are also worth trying.

**Break Symmetry:** The permutation symmetry of the hidden units makes the weights symmetrical, thus prone to co-adaptation in training. In DSD, pruning the weights breaks the symmetry of the hidden units associated with the weights, and the weights are asymmetrical in the final dense phase.

## 6  CONCLUSION

We introduce DSD, a dense-sparse-dense training framework that regularizes neural networks by pruning and then restoring connections. Our method learns which connections are important during the initial dense solution. Then it regularizes the network by pruning the unimportant connections and retraining to a sparser and more robust solution with same or better accuracy. Finally, the pruned connections are restored and the entire network is retrained again. This increases the dimensionality of parameters, and thus model capacity, from the sparser model.

DSD training achieves superior optimization performance. We highlight our experiments using GoogLeNet, VGGNet, and ResNet on ImageNet; NeuralTalk on Flickr-8K; and DeepSpeech-1&2 on the WSJ dataset. This shows that the accuracy of CNNs, RNNs, and LSTMs can be significantly improved with DSD training. Our numerical results and empirical tests show the inadequacy of current training methods for which we have provided an effective solution.

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

## A. APPENDIX: SIGNIFICANCE OF DSD IMPROVEMENTS

DSD training improves the baseline model performance by consecutively pruning and re-densifying the network weights. We conducted more intensive experiments to validate that the improvements are significant and not due to any randomness in the optimization. In order to evaluate the significance, we repeated the baseline training, DSD training (retraining on baseline) and conventional fine-tuning (retraining on the same baseline) multiple times. The statistical significance of DSD improvements are quantified on the Cifar-10 dataset using ResNet.

### 1. SIGNIFICANT IMPROVEMENTS ON CIFAR-10 USING RESNET-20

Cifar-10 is a smaller image recognition benchmark with 50,000 32x32 color images for training and 10,000 for testing. Training on Cifar-10 is fast enough that it is feasible to conduct intensive experiments within a reasonable time to evaluate DSD performance. The baseline models were trained with the standard 164 epochs and initial LR of 0.1 as recommended in the released code (Facebook, 2016). After 164 epochs, we obtained the model with a 8.26% top-1 testing error that is consistent with the Facebook result. Initialized from this baseline model, we repeated 16 times of re-training using DSD training and 16 times using conventional fine-tuning. The DSD used sparsity of 50% and 90 epochs (45 for sparse training and 45 for re-densing training). As a fair comparison, the conventional fine-tuning is also based on the *same* baseline model with the *same* hyper-parameters and settings (90 epochs, 45 LR of 0.001 and 45 LR of 0.0001).

Detailed results are listed below. On Cifar-10 and using ResNet-20 architecture, the DSD training on average achieved Top-1 testing error of 7.89%, which is a 0.37% absolute improvement (4.5% relative improvement) over the baseline model and relatively 1.1% better than the conventional fine-tuning. The experiment also shows that DSD training can reduce the variance of learning: the trained models after the sparse training and the final DSD training both have lower standard deviation of errors compared with their counterparts using conventional fine-tuning.

Table 10: Validation of DSD on Cifar10 data using ResNet-20

| ResNet-20 | Avg. Top-1 Err | SD. Top-1 Err | Sparsity | Epochs | LR |
|---|---|---|---|---|---|
| Baseline | 8.26% | - | 0% | 164 | 1e-1 |
| Direct Finetune (First half) | 8.16% | 0.08% | 0% | 45 | 1e-3 |
| Direct Finetune (Second half) | 7.97% | 0.04% | 0% | 45 | 1e-4 |
| DSD (Fist half, Sparse) | 8.12% | 0.05% | 50% | 45 | 1e-3 |
| DSD (Second half, Dense) | **7.89%** | **0.03%** | 0% | 45 | 1e-4 |
| Improve from baseline(abs) | 0.37% | - | - | - | - |
| Improve from baseline(rel) | **4.5%** | - | - | - | - |

We used t-test (unpaired) to compare the top-1 testing error rate of the models trained using DSD and conventional methods. The results demonstrate the DSD training achieves significant improvements from both the baseline model (p<0.001) and conventional fine tuning (p<0.001).

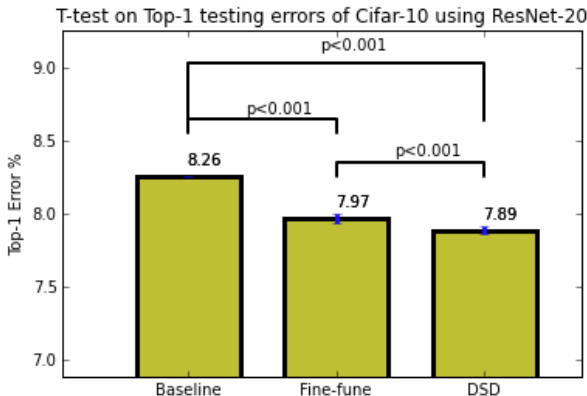

Figure 4: Significance of DSD improvements over baseline and fine-tune

Based on the results above, DSD significantly improves conventional baseline training and is also significantly better and more robust than conventional fine-tuning.

## B. Appendix: More Examples of DSD Training Improves the Captions Generated by NeuralTalk (Images from Flickr-8K Test Set)

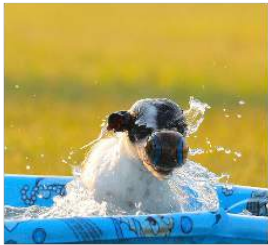
**Baseline**: a boy is swimming in a pool.
**Sparse**: a small black dog is jumping into a pool.
**DSD**: a black and white dog is swimming in a pool.

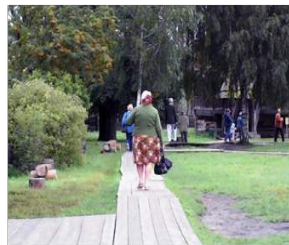
**Baseline**: a group of people are standing in front of a building.
**Sparse**: a group of people are standing in front of a building.
**DSD**: a group of people are walking in a park.

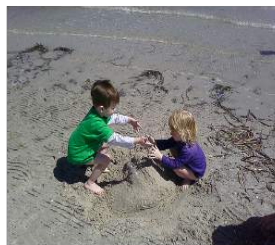
**Baseline**: two girls in bathing suits are playing in the water.
**Sparse**: two children are playing in the sand.
**DSD**: two children are playing in the sand.

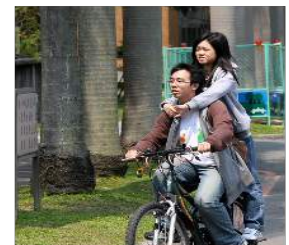
**Baseline**: a man in a red shirt and jeans is riding a bicycle down a street.
**Sparse**: a man in a red shirt and a woman in a wheelchair.
**DSD**: a man and a woman are riding on a street.

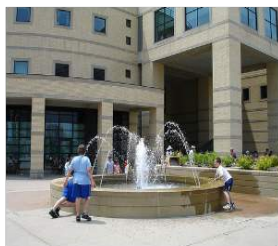
**Baseline**: a group of people sit on a bench in front of a building.
**Sparse**: a group of people are standing in front of a building.
**DSD**: a group of people are standing in a fountain.

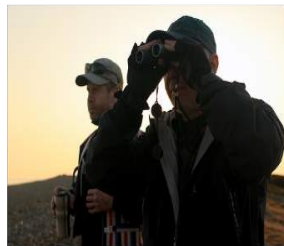
**Baseline**: a man in a black jacket and a black jacket is smiling.
**Sparse**: a man and a woman are standing in front of a mountain.
**DSD**: a man in a black jacket is standing next to a man in a black shirt.

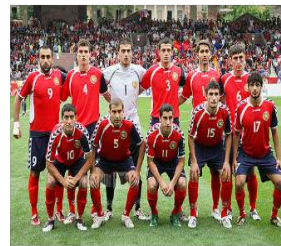
**Baseline**: a group of football players in red uniforms.
**Sparse**: a group of football players in a field.
**DSD**: a group of football players in red and white uniforms.

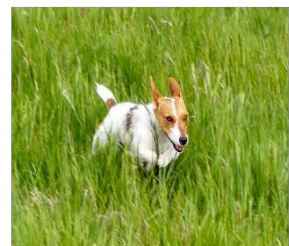
**Baseline**: a dog runs through the grass.
**Sparse**: a dog runs through the grass.
**DSD**: a white and brown dog is running through the grass.

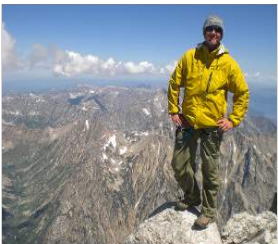
**Baseline**: a man in a red shirt is standing on a rock.
**Sparse**: a man in a red jacket is standing on a mountaintop.
**DSD**: a man is standing on a rock overlooking the mountains.

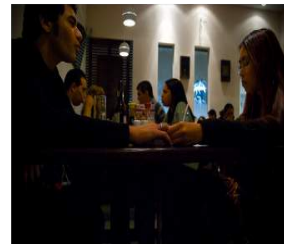
**Baseline**: a group of people are sitting in a subway station.
**Sparse**: a man and a woman are sitting on a couch.
**DSD**: a group of people are sitting at a table in a room.

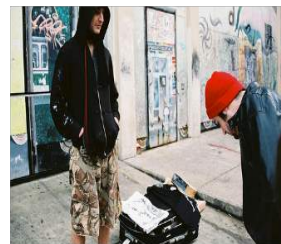
**Baseline**: a man in a red jacket is standing in front of a white building.
**Sparse**: a man in a black jacket is standing in front of a brick wall.
**DSD**: a man in a black jacket is standing in front of a white building.

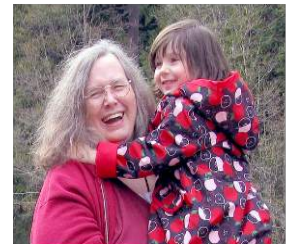
**Baseline**: a young girl in a red dress is holding a camera.
**Sparse**: a little girl in a pink dress is standing in front of a tree.
**DSD**: a little girl in a red dress is holding a red and white flowers.

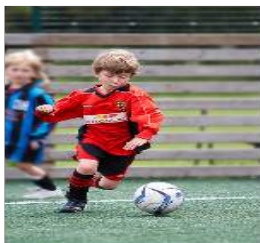
**Baseline**: a soccer player in a red and white uniform is playing with a soccer ball.
**Sparse**: two boys playing soccer.
**DSD**: two boys playing soccer.

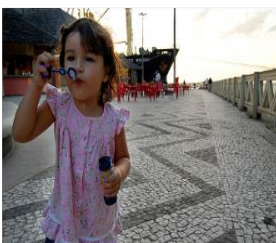
**Baseline**: a girl in a white dress is standing on a sidewalk.
**Sparse**: a girl in a pink shirt is standing in front of a white building.
**DSD**: a girl in a pink dress is walking on a sidewalk.

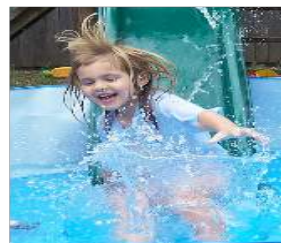
**Baseline**: a young girl in a swimming pool.
**Sparse**: a young boy in a swimming pool.
**DSD**: a girl in a pink bathing suit jumps into a pool.

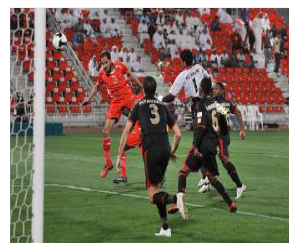
**Baseline**: a soccer player in a red and white uniform is running on the field.
**Sparse**: a soccer player in a red uniform is tackling another player in a white uniform.
**DSD**: a soccer player in a red uniform kicks a soccer ball.

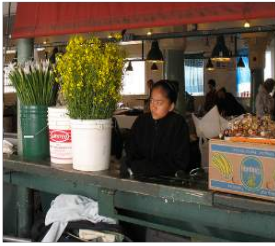

**Baseline**: a man in a red shirt is sitting in a subway station.
**Sparse**: a woman in a blue shirt is standing in front of a store.
**DSD**: a man in a black shirt is standing in front of a restaurant.

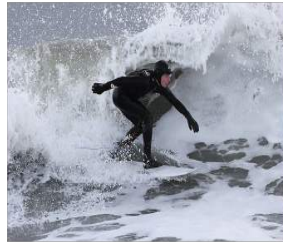

**Baseline**: a surfer is riding a wave.
**Sparse**: a man in a black wetsuit is surfing on a wave.
**DSD**: a man in a black wetsuit is surfing a wave.

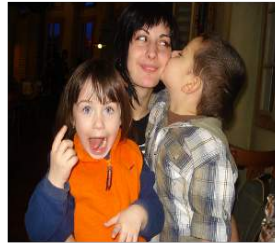

**Baseline**: two young girls are posing for a picture.
**Sparse**: a young girl with a blue shirt is blowing bubbles.
**DSD**: a young boy and a woman smile for the camera.

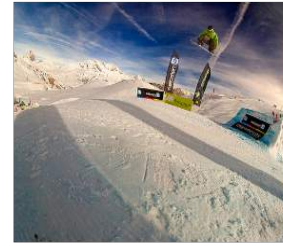

**Baseline**: a snowboarder flies through the air.
**Sparse**: a person is snowboarding down a snowy hill.
**DSD**: a person on a snowboard is jumping over a snowy hill.

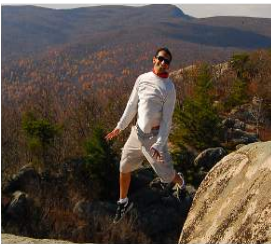

**Baseline**: a man in a red shirt is standing on top of a rock.
**Sparse**: a man in a red shirt is standing on a cliff overlooking the mountains.
**DSD**: a man is standing on a rock overlooking the mountains.

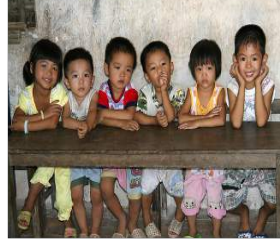

**Baseline**: a group of people sit on a bench.
**Sparse**: a group of people are sitting on a bench.
**DSD**: a group of children are sitting on a bench.

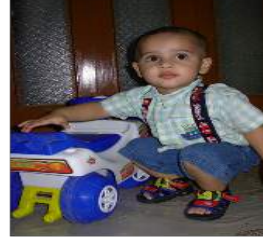

**Baseline**: a little boy is playing with a toy.
**Sparse**: a little boy in a blue shirt is playing with bubbles.
**DSD**: a baby in a blue shirt is playing with a toy.

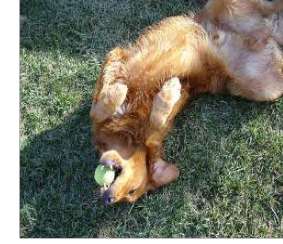

**Baseline**: a brown dog is running through the grassy.
**Sparse**: a brown dog is playing with a ball.
**DSD**: a brown dog is playing with a ball.

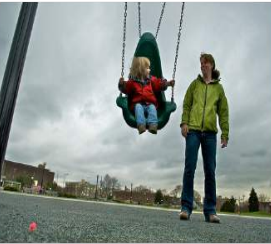

**Baseline**: a boy in a red shirt is jumping on a trampoline.
**Sparse**: a boy in a red shirt is jumping in the air.
**DSD**: a boy in a red shirt is jumping off a swing.

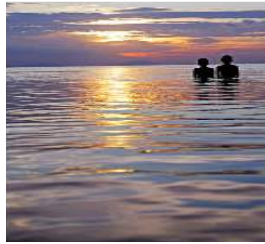

**Baseline**: a man is standing on the edge of a cliff.
**Sparse**: a man is standing on the shore of a lake.
**DSD**: a man is standing on the shore of the ocean.

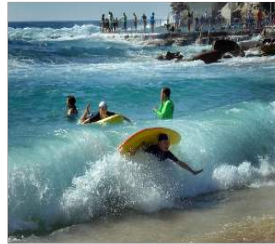

**Baseline**: two people are riding a boat on the beach.
**Sparse**: two people are riding a wave on a beach.
**DSD**: a man in a yellow kayak is riding a wave.

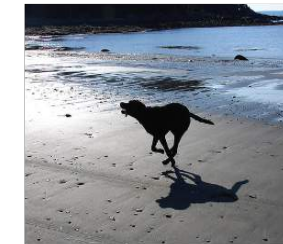

**Baseline**: a black and white dog is running on the beach.
**Sparse**: a black and white dog running on the beach.
**DSD**: a black dog is running on the beach.

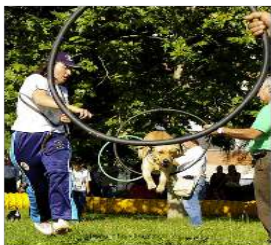

**Baseline**: a man and a dog are playing with a ball.
**Sparse**: a man and a woman are playing tug of war.
**DSD**: a man and a woman are playing with a dog.

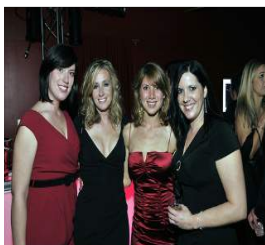

**Baseline**: a group of people are standing in a room.
**Sparse**: a group of people gather together.
**DSD**: a group of people are posing for a picture.

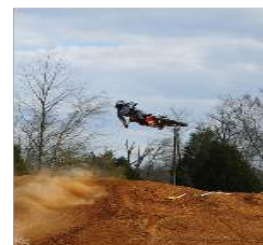

**Baseline**: a man in a red jacket is riding a bike through the woods.
**Sparse**: a man in a red jacket is doing a jump on a snowboard.
**DSD**: a person on a dirt bike jumps over a hill.

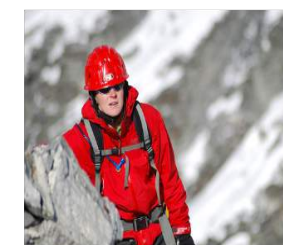

**Baseline**: a man in a red jacket and a helmet is standing in the snow.
**Sparse**: a man in a red jacket and a helmet is standing in the snow.
**DSD**: a man in a red jacket is standing in front of a snowy mountain.

