# Peer review of "DSD: Dense-Sparse-Dense Training for Deep Neural Networks"

_ICLR 2017 — accepted_

[Official Review · AnonReviewer1 · rating 5 · confidence 4 · 10 Dec 2016]
**Interesting training strategy for deep networks**

This paper presents a training strategy for deep networks.  First, the network is trained in a standard fashion.  Second, small magnitude weights are clamped to 0; the rest of the weights continue to be trained.  Finally, all the weights are again jointly trained.  Experiments on a variety of image, text, and speech datasets demonstrate the approach can obtain high-quality results.

The proposed idea is novel and interesting.  In a sense it is close to Dropout, though as noted in the paper the deterministic weight clamping method is different.

The main advantage of the proposed method is its simplicity.  Three hyper-parameters are needed: the number of weights to clamp to 0, and the numbers of epochs of training used in the first dense phase and the sparse phase.  Given these, it can be plugged in to training a range of networks, as shown in the experiments.

The concern I have is regarding the current empirical evaluation.  As noted in the question phase, it seems the baseline methods are not trained for as many epochs as the proposed method.  Standard tricks, such as dropping the learning rate upon "convergence" and continuing to learn, can be employed.  The response seems to indicate that these approaches can be effective.  I think a more thorough empirical analysis of performance over epochs, learning rates, etc. would strengthen the paper.  An exploration regarding the sparsity hyper-parameter would also be interesting.

[Official Review · AnonReviewer3 · rating 8 · confidence 3 · 15 Dec 2016 (modified: 20 Jan 2017)]
**models has the capacity to achieve higher accuracy with better training methods**

Summary: 
The paper proposes a model training strategy to achieve higher accuracy. The issue is train a too large model and you going to over-fit and your model will capture noise. Prune models or make it too small then it will miss important connections and under-fit. Thus, the proposed method involves various training steps: first they train a dense network, then prune it making it sparse then train a sparse network and finally they add connections back and train the model as dense again (DSD). The DSD method is generic method that can be used in CNN/RNN/LSTM. The reasons why models have better accuracy after DSD are: escape of saddle point, sparsity makes model more robust to noise and symmetry break allowing richer representations.

Pro:
The main point that this paper wants to show is that a model has the capacity to achieve higher accuracy, because it was shown that it is possible to compress a model without losing accuracy. And lossless compression means that there’s significant redundancy in the models that were trained using current training methods. This is an important observation that large models can get better accuracies as better training schemes are used. 

Cons & Questions:
The issue is that the accuracy is slightly increased (2 or 3%) for most models. And the question is what is the price paid for this improvement? Resource and performance concerns arises because training a large model is computationally expensive (hours or even days using high performance GPUs).

Second question, can I keep adding Dense, Sparse and Dense training iterations to get higher and higher accuracy improvement? Are there limitations to this DSDSD… approach?

[Official Review · AnonReviewer2 · rating 8 · confidence 3 · 20 Dec 2016]
**nice new training method for deep networks**

Training highly non-convex deep neural networks is a very important practical problem, and this paper provides a great exploration of an interesting new idea for more effective training.  The empirical evaluation both in the paper itself and in the authors’ comments during discussion convincingly demonstrates that the method achieves consistent improvements in accuracy across multiple architectures, tasks and datasets. The algorithm is very simple (alternating between training the full dense network and a sparse version of it), which is actually a positive since that means it may get adapted in practice by the research community.

The paper should be revised to incorporate the additional experiments and comments from the discussion, particularly the accuracy comparisons with the same number of epochs.

[Final Decision · Program Chairs · 06 Feb 2017]
**ICLR committee final decision**

Important problem, simple (in a positive way) idea, broad experimental evaluation; all reviewers recommend accepting the paper, and the AC agrees. Please incorporate any remaining reviewer feedback.